# Experimental Study on Enhanced Oil Recovery of Adaptive System after Polymer Flooding

**DOI:** 10.3390/polym15173523

**Published:** 2023-08-24

**Authors:** Yanfu Pi, Xinyu Fan, Li Liu, Mingjia Zhao, Linxiao Jiang, Guoyu Cheng

**Affiliations:** Key Laboratory of Enhanced Oil and Gas Recovery of Ministry of Education, Northeast Petroleum University, Daqing 163319, China; piyanfu@163.com (Y.P.); liuliduoduo@163.com (L.L.); zhaomingjia15@163.com (M.Z.); jianglinxiao6885@163.com (L.J.); zscgydyx@163.com (G.C.)

**Keywords:** after polymer flooding, adaptive system, preformed particle gel, a four-layer heterogeneous physical model, enhanced oil recovery

## Abstract

After polymer flooding in Daqing Oilfield, the heterogeneity of the reservoir is enhanced, leading to the development of the dominant percolation channels, a significant issue with inefficient circulation, a substantial amount of displacement agents, and elevated cost. In order to further improve oil recovery, an adaptive oil displacement system (ASP-PPG) was proposed by combining preformed particle gel (PPG) with an alkali-surfactant-polymer system (ASP). This comprehensive study aims to assess the effectiveness of the adaptive oil displacement system (ASP-PPG) in improving the recovery efficiency of heterogeneous reservoirs after polymer flooding. The evaluation encompasses various critical aspects, including static performance tests, flow experiments, microscopic experiments, profile control experiments, and flooding experiments conducted on a four-layer heterogeneous physical model. The experimental results show that the adaptive system has robust stability, enhanced mobility, effective plugging capability, and profile improvement capability. Notably, the system demonstrates the remarkable ability to successfully pass through the core and effectively block the large pores, resulting in an 18.4% recovery incremental after polymer flooding. This improvement is reflected in the reduced oil saturation values in the ultra-high permeability, high permeability, medium, and low permeability layers, which are 5.09%, 7.01%, 13.81%, and 15.45%, respectively. The adaptive system effectively recovered the remaining oil in the low and medium permeability layers, providing a promising approach for improving the recovery factors under challenging reservoir conditions.

## 1. Introduction

In recent years, with the development of petroleum exploration and exploitation in China, many old oil fields in the East have conducted water flooding and transited to a later phase, characterized by high water cut and high recovery factors. The traditional water flooding method is inadequate to sustain the high-efficiency production demanded by oilfields. Consequently, chemical flooding as a secondary oil recovery technology has gradually become the primary enhanced oil recovery method and has been applied to old oilfields after water flooding. For instance, polymer flooding technology has been widely utilized in Daqing Oilfield since 1996, resulting in remarkable economic effects and confirming the feasibility of the technique in enhancing oil recovery in reservoirs after water flooding [1,2,3]. However, the oil reservoirs exhibit pronounced heterogeneity in both the horizontal and vertical directions after polymer flooding, leading to a significant challenge in profile control [4,5]. In addition, the remaining oil after polymer flooding is highly scattered. It is urgent to develop novel test methodologies to improve the efficiency of chemical flooding for thick layers after polymer flooding.

Numerous endeavors have been undertaken to address the issues of highly heterogeneous reservoirs and complex dispersed remaining oil after polymer flooding. Xu et al. [6] conducted a parallel tube chamber flow experiment, achieving effective control over the water absorption capacity of the high permeability layer by crosslinking a high-concentration polymer solution with a fixative. Nie et al. [7] used polymer microspheres to block the pore throats, resulting in enhancing the permeability of the high permeability layer and facilitating successful profile control. Yin et al. [8] conducted enhanced oil recovery experiments after polymer flooding using a ternary composite system comprised of a polymer solution and a weak alkali-surfactant-polymer agent. Zhu et al. [9] conducted an indoor oil displacement experiment utilizing a composite system dominated by petroleum sulfonate, thus optimizing the oil displacement system to achieve the optimal effects. Yernazarova et al. [10] evaluated the feasibility of microbial flooding after polymer flooding through indoor physical simulation experiments. They found that microbial flooding had a higher potential for enhanced oil recovery of the reservoir after polymer flooding. In addition, Wu et al. [11] carried out oil displacement experiments using a large-scale three-dimensional physical heterogeneous model. They proposed the method of combining the profile control method, high-viscosity polymer flooding, and ternary composite flooding. High-viscosity polymer flooding increases the swept volume, while ternary composite flooding utilizes the remaining oil in the medium and low permeability layers. Thus, the proposed method realizes the purpose of improving recovery after polymer flooding. Khan et al. [12] designed functional plugs with different viscosity and physical properties and applied them to the ternary composite flooding after polymer flooding. This approach remarkably improved the recovery efficiency of the medium and low permeability layers while extracting the maximum potential from the residual oil within these layers. Liu et al. [13] prepared a binary compound oil expulsion system using surfactants with low interfacial tension and partially hydrolyzed polyacrylamide, significantly improving the recovery factors of both the high and low permeability layers. Cheraghian et al. [14] developed a novel nano-polymer oil displacement system that combined nanoparticle TiO_2_ with a polymer form, subsequently conducting indoor heterogeneous core oil displacement experiments. This research highlighted the favorable flow performance of nano-polymers in core displacement tests, yielding a 4% increase in recovery compared to the polymer solution.

Based on the research of relevant scholars, field tests were conducted after polymer flooding in Daqing Oilfield using high-concentration polymer flooding and weak alkali ternary composite flooding, which increased the recovery factor by 8% and 10%, respectively. However, the reservoir heterogeneity was intensified and caused the following issues: developing dominant percolation channels, inefficient and ineffective circulation, excessive displacement agent usage, and high costs [15,16].

To address these critical concerns, scholars have proposed the concept of utilizing pre-crosslinked particles (PPG) as a deep fluid flow diversion agent to block the dominant seepage channel and enhance the swept volume. Bai et al. [17] indicated that PPG particles expand after water absorption, possessing specific viscoelastic properties, compressive strength, and water retention effects after expansion. These particles offer the distinct advantages of simple preparation, convenient construction, decent temperature and salt resistance, and a small amount of pollution to the reservoir. Lenji et al. [18] studied the impact of water salinity on PPG water absorption and expansion, identifying a decreasing trend in PPG’s water absorption and expansion performance with the salinity increase. Lai et al. [19] studied the migration characteristics and profile control ability of PPG particles in porous media through laboratory tests. They claimed that PPG particles migrate in porous media in the pattern of “aggregation blockage—pressure increase—deformation passage—pressure decrease”, indicating an effective profile improvement capability. Wang et al. [20] optimized the concentration of each component in PPG and the ternary system through laboratory experiments. Alhuraishawy A. K. et al. [21] conducted experiments to evaluate the performance of combining PPG with low salinity water flooding in improving the recovery efficiency of fractured carbonate reservoirs under laboratory experimental conditions. The results revealed that the synergistic effect of PPG and low salinity water flooding could effectively improve oil recovery. Sang et al. [22] emphasized that PPG could effectively improve reservoir heterogeneity and the recovery factor within the low permeability layer. Notably, they highlighted that the effectiveness of PPG in improving recovery within low permeability layers was more pronounced in reservoirs with greater heterogeneity. Gao et al. [23] systematically investigated the profile control effect of the PPG/ASP system in heterogeneous reservoirs through a series of experiments. They claimed that the PPG/ASP system was superior to the ASP single system in profile control and enhanced oil recovery after polymer flooding. Compared to the pure ASP system, the PPG/ASP system had greater drag and residual drag coefficients, higher viscosity, and better flow stability. Gong et al. [24] studied the enhanced oil recovery effect of the B-PPG/HPAM system in heterogeneous porous media through indoor core displacement experiments, indicating that the B-PPG/HPAM system had a higher recovery efficiency than the HPAM or B-PPG system alone. The efficacy of the B-PPG system was attributed to its unique characteristics, including blocking, deformation, and passing through the pore during the flow process. This enabled the system to regulate the flow across different permeability layers, leading to a higher drag coefficient and residual drag coefficient than the single HPAM or B-PPG system. Wu et al. [25] evaluated the profile control effects of the ASP-PPG system and the ASP system under different permeability levels through laboratory experiments. Their findings revealed that, even when the permeability level difference exceeded 30, the ASP-PPG system retained the capacity to effectively block the dominant seepage channels, highlighting its robust profile control capabilities under challenging reservoir conditions.

The development and application of PPG provide a novel approach to developing enhanced oil recovery technology. In Daqing Oilfield, the innovative PPG particles developed by the Daqing Exploration and Development Research Institute have been synergistically integrated into a ternary (polymer, surfactant, and alkali) composite system to form an adaptive oil displacement system with promising practical applications. However, the scholars conducted laboratory experiments primarily relying on the one-dimensional physical model, which cannot accurately simulate the actual reservoir and the heterogeneity of the reservoir. Therefore, we developed a visual oil saturation monitoring apparatus, along with a sophisticated three-dimensional four-layer heterogeneous core well pattern model. This comprehensive setup can observe the changes in oil saturation in real time and accurately simulate the heterogeneity between and within the actual reservoir.

This study evaluated the static performance (viscoelasticity and stability) of the adaptive system and performed multiple tests, including a one-dimensional long homogeneous core flood experiment, a microscopic visualization model (lithography glass) experiment, a two-pipe parallel experiment, and a four-layer heterogeneous well pattern model experiment. In addition, we investigated the flow and transportation behaviors of PPG particles in the adaptive system and its plugging profile control performance. The displacement effect of the adaptive system was quantitatively evaluated. The findings of this research provide a theoretical basis for the application of the adaptive system flooding after polymer flooding in Daqing Oilfield.

## 2. Materials and Methods

### 2.1. Experimental Materials

The chemical and experimental materials used in this study are presented as follows:

Polymer: Partially hydrolyzed polyacrylamide (HPAM) with a molecular weight of 15 million and 25 million, effective content of 90%, and hydrolysis degree of 30%. This specific polymer was produced by the Daqing Refining and Chemical Company (Daqing, China).

Surface active agent: Petroleum sulfonate, mass concentration 20%, produced by Daqing Wantong Chemical Co., Ltd. (Daqing, China).

Alkali: Purity 99% anhydrous Na_2_CO_3_, provided by the Exploration and Development Research Institute of Daqing Oilfield (Daqing, China).

Preformed particle gel (PPG): The size of the synthesized PPG particles was 0.15–0.3 mm. Under the experimental temperature of 45 °C, the PPG particles expanded through water absorption. The final stable expansion factor was 3, and the compressive strength was 1.7 MPa. The PPG particles were developed by the Research Institute of Exploration and Development of Daqing Oilfield (Daqing, China).

Water: Produced water and clean water provided by the first oil production plant of Daqing Oilfield. The simulated formation water prepared in the laboratory had a salinity of 6778 mg/L, and the dosage of each reagent is shown in Table 1.

Experimental oil: The simulated oil was mixed with crude oil and aviation kerosene from the first oil production plant of Daqing Oilfield. The measured viscosity was 9.8 mPa·s at 45 °C.

The component concentrations of the adaptive system (ASP-PPG) used in the experiment were 400 mg/L for PPG, 1200 mg/L for the polymer, 1.2% for the alkali, and 0.2% for the surfactant. The ASP system used in the static performance evaluation experiment had the same concentration as the ASP-PPG system.

A comprehensive analysis of the permeability distribution was conducted based on 20 coring wells located in various areas of Daqing Oilfield, including Lamadian, Sazhong, Sanan, and Sabei. According to the statistical results of sealed cores from the wells after polymer flooding in the test area of Daqing Oilfield, the parameters of the laboratory physical model were determined. The permeability of the low permeability, medium, high, and extra-high permeability layers was approximately 200 × 10^−3^ μm^2^, 500 × 10^−3^ μm^2^, 1800 × 10^−3^ μm^2^, and 3000 × 10^−3^ μm^2^, respectively. Additionally, a hyperosmotic strip with a permeability of approximately 5000 × 10^−3^ μm^2^ was implemented in the ultra-high permeability layer, considering the tendency for the actual reservoir to form high permeability bands after polymer flooding. With these considerations, our indoor experimental physical model closely simulated real-world reservoir conditions.

The one-dimensional long homogeneous core model, the microscopic lithography glass model, the two-pipe parallel core model, and the four-layer heterogeneous core pattern model were designed for the seepage performance evaluation of the adaptive system, the microscopic migration of the adaptive system in porous media, the profile control performance evaluation, and the oil displacement effect evaluation of the adaptive system, respectively.

Among them, the one-dimensional long homogeneous core model is also called the porous pressure measurement seepage model. Four pressure measurement points are equidistantly located in the model. The pressure measurement point closest to the injection end is pressure measurement point 1, followed by pressure measurement points 2, 3, and 4, which are used to observe the pressure change. The specific parameters of the model are shown in Table 2. In the microscopic lithography glass model, the parameters are 63 × 63 mm, internal model size 40 × 40 mm. In the two-pipe parallel model, two different permeability casting thin cores are selected for the combination, and the parameters of the model are shown in Table 3. In the four-layer heterogeneous core well pattern model, 36 pairs of electrodes are arranged in each layer. The saturation monitoring device can dynamically monitor the change in oil saturation for each model layer by measuring the resistance between the electrode pairs, and nine pressure measurement points are arranged on it to observe the pressure distribution. The specific parameters are listed in Table 4.

### 2.2. System Static Performance Evaluation Experiment

The PPG solution was prepared with clear water, enabling the PPG particles to adsorb water and expand. Additionally, the ternary system (ASP) was prepared by the method of clear water preparation and the produced water dilution. Then, the PPG and ASP solutions were mixed, and the DJ1C-90 augmented electric mixer was used to stir the mixture to achieve a dispersed state and obtain the adaptive system (ASP-PPG). The Physica MCR 301 rotational rheometer was used to measure the elastic modulus G′ and viscous modulus G″ of the ASP-PPG system and the ASP system, respectively. These two parameters, G′ and G″, were utilized to characterize the viscoelasticity of the system. Next, the prepared adaptive system solution was placed in a 45 °C incubator for 90 days. The viscosity of the ASP-PPG system was measured using a DV-II+Pro digital display viscometer (Brookfield, WI, USA) at certain intervals. Additionally, the interfacial tension between the ASP-PPG system and the simulated oil at different contact times was measured by a TX500 interface tensiometer (Cono, Elizabeth, NJ, USA), providing insights into the stability of the adaptive system.

### 2.3. Flow Experiment

In order to evaluate the mobility of the adaptive system, a one-dimensional long core flow experiment was carried out. In this experiment, the adaptive system (ASP-PPG) was injected into the core using an ISCO pump, and the PPG particles were mainly injected into the core through the carrying effect of the polymer solution. The schematic of the experimental device is shown in Figure 1. The experimental procedure of the experiment is presented as follows: (1) The 1000 mm long core was evacuated and saturated with water. (2) The simulated formation water was then injected into the core at a flow rate of 0.5 mL/min until achieving a stable pressure at the injection end. (3) The injection speed maintained constant as the adaptive system was injected into the core until pressure equilibrium was achieved at the injection end. (4) The flow rate remained, and the subsequent stage witnessed water flooding until pressure stabilization at the injection end, thereby concluding the experiment. The pressure values of each stage were recorded every 20 min, allowing the calculation of the drag coefficient *F_R_* and residual drag coefficient *F_RR_*. The drag coefficient and residual drag coefficient were determined by the following equations [26].
(1)$$F_{R}=\Delta P_{2}/\Delta P_{1}$$
(2)$$F_{RR}=\Delta P_{3}/\Delta P_{1}$$
where *F_R_* is the resistance coefficient of the adaptive system, *F_RR_* is the residual drag coefficient of the adaptive system, and ∆*P*_1_ is the pressure difference between the two ends of the core in the stationary phase of water flooding. ∆*P*_2_ is the pressure difference between the two ends of the core in the stationary phase of the adaptive system flooding. ∆*P*_3_ is the pressure difference between the two ends of the core at the stationary stage of the subsequent water flooding.

### 2.4. Experiment on the Microscopic Transport Behavior of the System

In this study, the microscopic visualization model (lithography glass) was used to study the deformation of PPG particles as they passed through the pore throats. Additionally, we delved into the microscopic transport behavior of the adaptive system in the lithography glass. The schematic of the experimental apparatus is presented in Figure 2. The specific experimental steps were as follows: (1) The microscopic lithography glass model was evacuated, saturated with water, and then saturated with simulated oil. (2) The flow rate was set to 0.003 mL/min, and the simulated formation water was injected into the microlithography glass to reach a content of 98%. (3) The injection rate was maintained, and the adaptive system solution was injected into the lithography glass, and the experiment ended. During the experiment, we closely observed the dynamic process of the remaining oil at the same positions and PPG behavior as it passed pore throats and recorded the displacement process images.

### 2.5. Profile Control Evaluation Experiment

The profile control performance of the ASP-PPG system is reflected by the adaptive system’s profile improvement ability. Four groups of two-pipe parallel experiments were performed using different permeability levels. These experiments encompassed combinations involving the combination of the ultra-high permeability layer and high permeability layer, as well as the combination of the medium permeability layer and low permeability layer. The schematic of the experimental setup is shown in Figure 3. The experimental procedure is delineated as follows: (1) The casting core was evacuated and saturated with water. (2) The flow rate was set to 0.5 mL/min, and the simulated formation water was injected into the core to reach the content of 98%. (3) The 0.5 PV polymer solution was injected with a constant flow rate, driving the water content up to 98%. (4) The 0.7 PV of the adaptive system was pumped into the core at the same flow rate. (5) The experiment was ended by maintaining an unaltered injection speed, witnessing subsequent water flooding until the moisture content reached 98%. During these experiments, the pressure at the injection end and the production of the high and low permeability layers were recorded every 10 min. These recorded data enabled the calculation of the diversion rate and the profile improvement rate *f*. The profile improvement rate was determined by the following equation [27]:(3)$$f=\frac{Q_{1}/q_{1}-Q_{2}/q_{2}}{Q_{1}/q_{1}},$$
where *f* is the profile improvement rate, *Q*_1_ is the amount of liquid absorption before profile control of the hyperosmotic layer, *Q*_2_ is the amount of liquid absorption after profile control in the hyperosmotic layer, *q*_1_ is the amount of liquid absorption before profile control of low permeability layer, and *q*_2_ is the amount of liquid absorption after profile control in the low permeability layer.

### 2.6. Experiment on the Oil Displacement Effect Evaluation of the Adaptive System

According to the actual conditions of the reservoir, this experiment used the configuration of a two-injection and two-production well pattern for evaluation. The core displacement experiments were performed on a four-layer heterogeneous well pattern to investigate the displacement effect of the adaptive system. The schematic representation of the experimental apparatus is shown in Figure 4. The specific steps are as follows: (1) The four-layer heterogeneous core well pattern model was evacuated and saturated with water and oil. Then, the formation water was injected into the core at a flow rate of 4 mL/min using one injection and one production approach to reach a 98% water content. (2) The 0.57 PV (1000 mg/L) medium polymer solution was injected into the core at a constant rate and then conducted subsequent water flooding to reach a 98% water content. (3) With the same injection rate, 0.5 PV of the adaptive system solution was injected into the core using two injections and two production approaches. Then, 0.2 PV of a polymer slug with equal viscosity was injected into the core. Finally, the experiment concluded with subsequent water flooding until achieving a 98% water content.

Throughout the experiment, the pressure, liquid production, oil production, and water production of the experiment were recorded to estimate the recovery and water cut of each stage. The saturation monitoring system was used to record the oil saturation distribution of different permeability reservoirs in each displacement stage, creating the saturation distribution map.

The oil saturation detection device was used to measure the oil saturation of the core through the rock electric experiment method (electrode method). The rock electric theory (electrode method) was based on Archie’s formula, a fundamental relationship capturing the interplay between rock electric parameters, such as resistivity, porosity, and water saturation. Archie’s equations are presented as follows [28,29,30]:(4)$$F=\frac{R_{o}}{R_{w}}=\frac{a}{\varphi^{m}},$$
(5)$$I=\frac{R_{t}}{R_{o}}=\frac{b}{S_{w}^{n}},$$
where *F* is the relative resistivity, *R_o_* is the rock resistivity of 100% saturated formation water, and *R_w_* is the resistivity of formation water in rock pores. *φ* is the porosity, and m is the cementation index related to the pore structure and cementation degree. *I* is the increased resistance rate, *R_t_* is the resistivity of oil-bearing rock, n is the stratum index, *S_w_* is water saturation, So is oil saturation, and a and b are the lithologic constants related to rock properties.

For the four-layer heterogeneous model with uniform thickness, the real-time resistance values can be measured by the resistance value measurement device at different displacement stages. By combining the resistivity–oil saturation relationship curve under the given permeability, the oil saturation distribution map was generated for each layer within the complete four-layer heterogeneous core pattern model.

## 3. Results and Discussion

### 3.1. Static Performance Analysis of the System

The viscoelasticity of the system is an important parameter reflecting the blocking and profile control ability of the flooding system. The viscoelasticity was quantified using two key metrics: a storage modulus and loss modulus. G′ is the storage modulus (also known as elastic modulus), indicating the strength of the system. G″ is the loss modulus (also called viscous modulus), which represents the viscosity of the system. The change of the moduli (G′ and G″) of the ASP-PPG system and the ASP system under different oscillation frequencies was tested experimentally, and the results are shown in Figure 5. Under the condition of the same oscillation frequency and the same concentration, the values of the elastic modulus G′ and viscous modulus G″ of the ASP-PPG system are greater than those of the single ASP system, which indicates that the preformed particle gel (PPG) can increase the viscoelasticity of the ASP system.

The long-term stability of the adaptive system in the oilfield site directly affects the production cost. The distance between the injection and production wells in the field exceeds 100 m. This configuration demands prolonged injection times, making robust system stability a fundamental necessity from an engineering perspective. In this study, the stability of the adaptive system was evaluated by testing the system viscosity and the oil–water interface tension at different times during the experiment, and the results are presented in Figure 6. As shown in Figure 6, the viscosity of the adaptive system is characterized by a modest initial increase, followed by a decrease, ultimately converging into a stable state. The final viscosity stabilizes at approximately 96 mPa·s, and the viscosity retention rate is over 90%. The oil–water interfacial tension of the adaptive system fails to show an apparent change trend with the increase in time. The interfacial tension is maintained at an ultra-low (10^−3^ mN/m) order of magnitude. Therefore, the adaptive system has robust stability and is conducive to long-term function in the field.

### 3.2. Evaluation of the Seepage Capacity

A comprehensive evaluation of the seepage performance is the key to investing in the migration of the adaptive system in porous media. Due to the poor injectability of PPG particles in 200 × 10^−3^ μm^2^ and 500 × 10^−3^ μm^2^ cores, 1800 × 10^−3^ μm^2^ and 3000 × 10^−3^ μm^2^ porous cores were selected in this paper. In the case of unsaturated crude oil formation, an adaptive system flow experiment was performed to observe the pressure changes at each pressure measurement point in different periods, and the results are shown in Figure 7. The pressure of the two cores at each stage of stability was recorded. In addition, the resistance coefficient F_R_ and residual resistance coefficient F_RR_ were calculated, and the results are listed in Table 5.

Figure 7 indicates that the injection end pressure within both permeability cores remains low during the water flooding stage, stabilizing at approximately 0.005 MPa. However, during the adaptive flooding stage, the pressure at the injection end of the core model with a permeability of 1800 × 10^−3^ μm^2^ rises rapidly. At around 2.3 PV of the cumulative injected pore volume, the pressure stabilizes at approximately 0.85 MPa, and the drag coefficient is 141.7. Then, a fluctuating pressure pattern with a noticeable zigzag shape occurs, indicating a distinctive migration process for PPG particles within the core’s pore throats as accumulation, pressure increase, deformation passage, and eventual pressure decrease. The pressure of the four pressure measurement points along the path sequentially increases, indicating a gradual migration of PPG particles into the core depth. Being closer to the injection end of the core results in an earlier starting time of the pressure surge and a larger stable pressure. Additionally, the resistance coefficients of each pressure point are 101.7, 68.3, 48.3, and 30.0, respectively, indicating that more PPG particles remain near the inlet, resulting in different blocking effects of each core section.

In the subsequent water flooding stage, the pressure at each measurement point fluctuates, declines with continuous water injection, and eventually becomes stable. However, the pressure at the stable stage is much greater than at the water flooding stage. The residual resistance coefficients of the injection end and the pressure measurement points along the path are 51.7, 40.0, 31.7, 26.7, and 20.0, respectively. This indicates that the adaptive system can successfully migrate to the depth of the core and can still block part of the pores in the subsequent water flooding stage with strong stability. Compared with the pressure curves of the core with a permeability of 3000 × 10^−3^ μm^2^, the pressure curves of the injection end and measurement points along the model have a similar trend but with different high-pressure ranges.

### 3.3. Microscopic Migration Behavior

The microscopic transport behavior of the adaptive system in porous media was studied using the microscopic lithography glass visualization model experiment, and the experimental results are shown in Figure 8. As shown in Figure 8, as the PPG particles block the pore throat, the PPG particles are deformed at the pore throat after water loss. If a substantial pressure differential exists and the compression speed significantly exceeds the speed of water loss and particle contraction, PPG is partitioned into smaller particles due to the large pressure difference and breaks through the pore throat. As shown by the red circle in the figure, PPG particles will be divided into smaller particles due to excessive pressure difference and then pass through the pore throat. Part of the PPG deforms and passes through the pore throat smoothly, while the other segment of the PPG enters the previously untouched regions due to the pore throat’s exerted normal force.

This breakthrough effect positions the PPG to access previously inaccessible regions, leveraging its elastic deformation capacity to drive oil displacement. The primary migration mode of PPG in the pore throat is elastic deformation or partition—migration—re-elastic deformation or partition—deep migration, which reduces the formation permeability and guides the subsequent injection fluid constantly entering the pore throats that were previously difficult to access. Thus, the sweep coefficient is greatly improved. The adaptive system exhibits a dual mechanism of elastic displacement and depth displacement.

### 3.4. Profile Control Capability Analysis

The larger the permeability range leads to stronger heterogeneity between layers. A sufficient profile control effect of the adaptive system is required to significantly impact the recovery factor of the low permeability layer. Thus, the PPG particles in the adaptive system need to block the ultra-high permeability layer to achieve “blocking without death” and still be able to smoothly and slowly pass through the ultra-high permeability layer. The natural pressure distribution further augments the penetration of subsequent injection fluids into the low permeability layer. In this study, four sets of two-pipe parallel experiments were performed using cores with different permeability levels. The profile control ability of the adaptive system was investigated by evaluating the profile improvement ability of the system. The diversion rates of different permeability levels are shown in Figure 9. In addition, the profile improvement rate of the system is calculated under different permeability levels, and the results are presented in Table 6.

From Figure 9 and Table 6, when the permeability level difference is 15, the ASP-PPG system achieves a significant reduction in the flow rate of the high permeability layer from 97.3% to 65% while boosting the flow rate of the low permeability layer from 2.7% to 35%. For the case with the permeability ratio of 9, the flow rate of the high permeability layer decreases from 93% to 57%, and the flow rate of the low permeability layer increases from 7% to 43% during the injection of the ASP-PPG system. When the permeability difference ratio is 6, the injection of the ASP-PPG system decreases the shunt rate of the high permeability layer from 89% to 53% and increases the shunt rate of the low permeability layer from 11% to 47%. When the permeability level difference is 3.6, the diversion rate of the high permeability layer decreases from 84% to 38%, and the diversion rate of the low permeability layer increases from 16% to 62%. Therefore, the ASP-PPG system can effectively improve the liquid absorption profile of the heterogeneous reservoir after polymer flooding. The smaller the permeability difference, the more obvious the improvement effect of the profile. The profile improvement rates at permeability differences of 15, 9, 6, and 3.6 are 94.85%, 90.02%, 86.06 and 88.33%, respectively. All of them exceed the crucial threshold of 85%, indicating that the ASP-PPG system can effectively block the high permeability layer, increase the seepage resistance, and force the subsequent displacement fluid to pass through the low permeability layer, significantly increasing the oil recovery of the low permeability layer.

### 3.5. Displacement Experiment of the Four-Layer Heterogeneous Well Pattern Model

#### 3.5.1. Determination of the Resistance–Oil Saturation Standard Relation Curve

The assessment of the adaptive system’s behavior in porous media was conducted using the steady-state method. The measured data, including resistance and oil saturation, from the cores with the permeability of 200 × 10^−3^ μm^2^, 500 × 10^−3^ μm^2^, 1800 × 10^−3^ μm^2^, and 3000 × 10^−3^ μm^2^ were analyzed, and the standard relationship curves between resistance and oil saturation are presented in Figure 10.

#### 3.5.2. Oil Displacement Effect Evaluation and Analysis of the Adaptive System of the Two-Injection and Two-Production Pattern after Polymer Flooding

The displacement experiments of the four-layer heterogeneous model were performed, and the experimental results of each stage are shown in Table 7.

As shown in Table 7, the recovery factor of the four-layer heterogeneous core pattern model is 32.4% in the water flooding stage, 15.6% in the polymer flooding stage, and 18.4% in the adaptive flooding stage, resulting in a total recovery of 66.4%.

The curves of each parameter with the cumulative injected pore volume and the change of the planar saturation field in the adaptive flooding stage of the four-layer heterogeneous core well pattern model experiments are shown in the figure below.
(1)Dynamic production curve

As shown in Figure 11, the pressure at the injection end slightly increases after the water flooding starts. With the increase in the injection time, the pressure at the injection end tends to be stable and eventually stabilizes at 0.049 MPa. The crude oil recovery rises sharply and then tends to be smooth, and the water cut also increases. The water flooding stops when the water content reaches 98%. In the polymer flooding process, the pressure at the injection end increases sharply, and the water cut decreases dramatically. The maximum decrease in the water cut is 26.52%. When injecting the polymer solution, the injection end pressure gradually becomes stable, and the water content gradually increases.

In the adaptive flooding stage of the two-injection and two-production pattern after polymer flooding, the pressure at the injection end rises rapidly at first, then tends to be stable and fluctuates within a specific value range, and the highest pressure reaches 1.665 MPa. At this stage, the water cut decreases first and then increases, and the maximum decrease in the water cut is 23.16%. Due to the well pattern, the original diversion line becomes the main streamline, which can effectively recover the remaining oil in the area, greatly reducing the water cut. The recovery curve initially shows a gradual rise, followed by acceleration and eventual stabilization.
(2)The planar saturation field in the adaptive flooding stage

The entire process of the adaptive flooding stage is monitored through a real-time monitoring device. Considering the relationship curve between resistance and oil saturation, the oil saturation change pattern of the different permeability layers before and after the adaptive flooding is generated by surfer8 software (Golden Software, Golden, CO, USA).

As shown in Figure 12a,b, the average oil saturation of the ultra-high permeability layer before injecting the adaptive system is approximately 29.72%, and the average oil saturation after the injection is around 24.63%. The oil saturation of the permeable layer decreases by 5.09%. This decrease signifies an enhanced utilization of previously untapped oil reservoirs, particularly on both sides of the blind end (challenging to access areas).

Figure 12c,d illustrate that the average oil saturation of the high permeability layer before injection of the adaptive system is 35.87%, and the average oil saturation after injection of the adaptive system is 28.86%. The oil saturation of the permeable layer decreases by about 7.01, indicating effective utilization of the remaining oil reservoirs. The utilization of the remaining oil in this permeable layer is similar to that in the ultra-high permeability layer, and the oil saturation decline is higher than that in the ultra-high permeability layer.

In Figure 12e,f, the average oil saturation in the medium permeability layer before the injection of the adaptive system is 52.27%, and the average oil saturation after the injection of the adaptive system is 38.46%. The oil saturation of the permeability layer decreases by about 13.81%. Adaptive flooding leads to gradual oil distribution along the flow path from injection to production wells, with higher utilization near the injection wells.

As shown in Figure 12g,h, the average oil saturation of the low permeability layer before injection of the adaptive system is 66.51%, and the average oil saturation after injection of the adaptive system is 51.06%. The oil saturation in the permeable layer decreases by about 15.45%. The behavior in this layer closely resembles the medium permeability layer but with a higher sweep area, indicating even more significant utilization. 

## 4. Conclusions

In this study, we address the intricate challenge of enhancing the recovery efficiency in heterogeneous reservoirs after polymer flooding by exploring the feasibility of the combined ASP-PPG flooding system. Our investigation involves static performance assessments, flow experiments, microscopic experiments, profile control experiments, and four-layer heterogeneous physical model flooding experiments. The following noteworthy conclusions are drawn:
(1)Preformed particle gel (PPG) notably enhances the viscoelasticity within the ASP-PPG system, exhibiting a superior elastic modulus (G′) and viscous modulus (G′) compared to the standalone ASP system. The adaptive system also has strong stability, with the viscosity retention exceeding 90% over a long period. Notably, the oil–water interface tension remains consistently at an ultra-low magnitude (10^−3^ mN/m).(2)The adaptive system exhibits remarkable mobility and can be smoothly delivered deep into the core. The adaptive system’s blocking effect on the core increases the resistance of the seepage, contributing to a large resistance coefficient and residual resistance coefficient.(3)Preformed particle gel (PPG) can access previously inaccessible oil areas while exhibiting elastic deformation features during oil displacement. The primary mode of PPG transport within the pore throat involves elastic deformation or partition—transport—elastic deformation or partition—deep transport.(4)The adaptive system can effectively improve the liquid absorption profile of heterogeneous reservoirs after polymer flooding. Notably, the profile improvement effect becomes more pronounced with diminishing permeability differences. For instance, in cases of a permeability level difference of 3.6, the dispersion rate in the high permeability layer decreases from 84% to 38%, and the dispersion rate in the low permeability layer increases from 16% to 62%, leading to an overall profile improvement rate of 88.33%.(5)The incremental recovery of the adaptive flooding stage after polymer flooding is 18.4%. The oil saturation within the ultra-high, high, medium, and low permeability layers declines 5.09%, 7.01%, 13.81%, and 15.45%, respectively. The adaptive system can effectively access the remaining oil in the medium and low permeability layers, significantly enhancing the recovery factor in these regions.

## Figures and Tables

**Figure 1 polymers-15-03523-f001:**
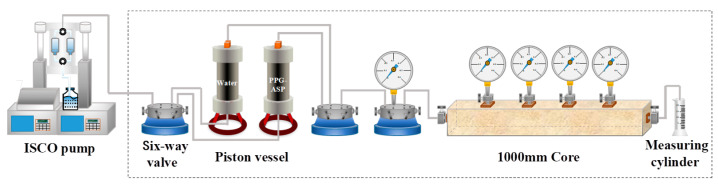
Schematic of the flow experimental device.

**Figure 2 polymers-15-03523-f002:**
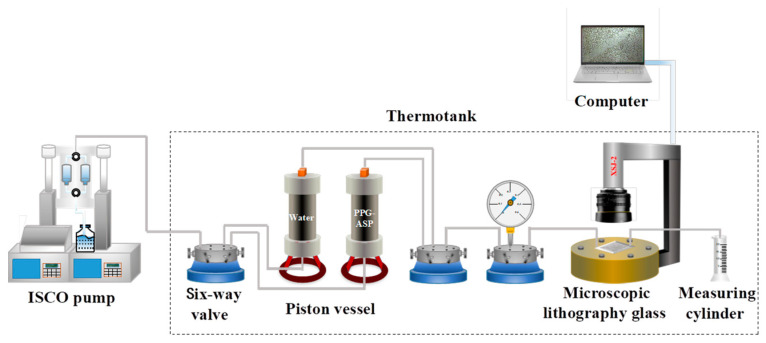
Schematic of the experimental apparatus for microscopic transport behavior.

**Figure 3 polymers-15-03523-f003:**
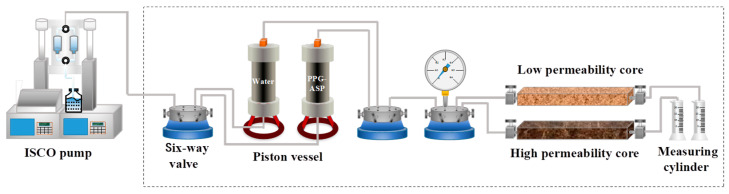
Schematic of the two-pipe parallel experimental setup.

**Figure 4 polymers-15-03523-f004:**
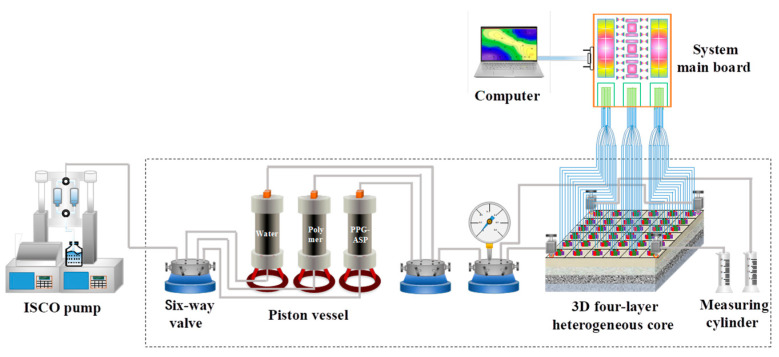
Schematic of the experiment for the four-layer heterogeneous model.

**Figure 5 polymers-15-03523-f005:**
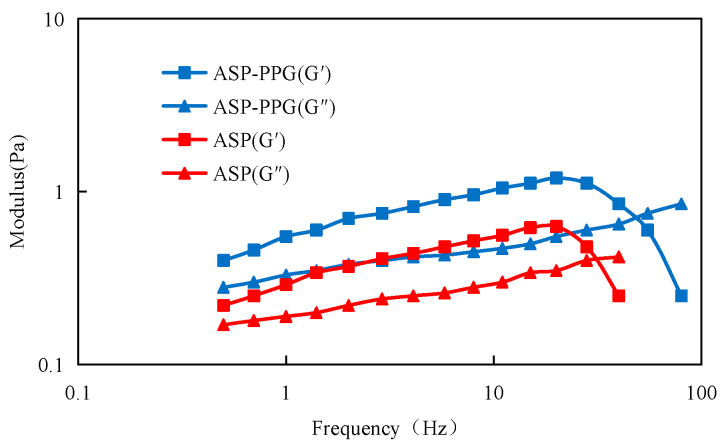
Viscoelastic curves of the ASP-PPG system and the ASP system.

**Figure 6 polymers-15-03523-f006:**
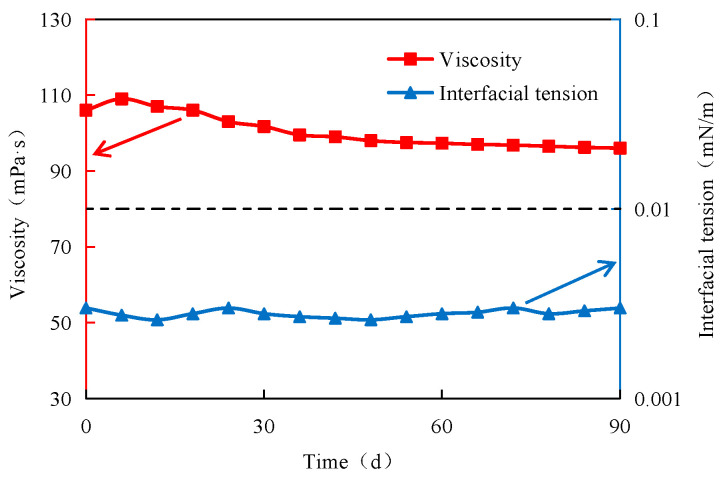
Viscosity and interfacial tension of the adaptive system as a function of time.

**Figure 7 polymers-15-03523-f007:**
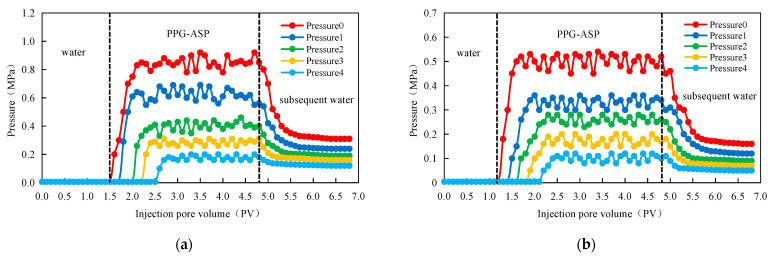
(**a**) Pressure variation curves for long core experiments with a permeability of 1800 × 10^−3^ μm^2^. (**b**) Pressure variation curves for long core experiments with permeability of 3000 × 10^−3^ μm^2^.

**Figure 8 polymers-15-03523-f008:**
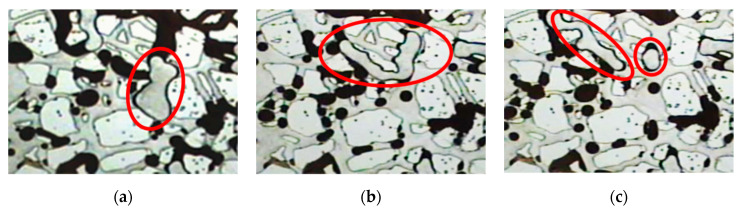
Channel mode of PPG in the pore throats (**a**) before the deformation, (**b**) during the deformation, and (**c**) after the deformation.

**Figure 9 polymers-15-03523-f009:**
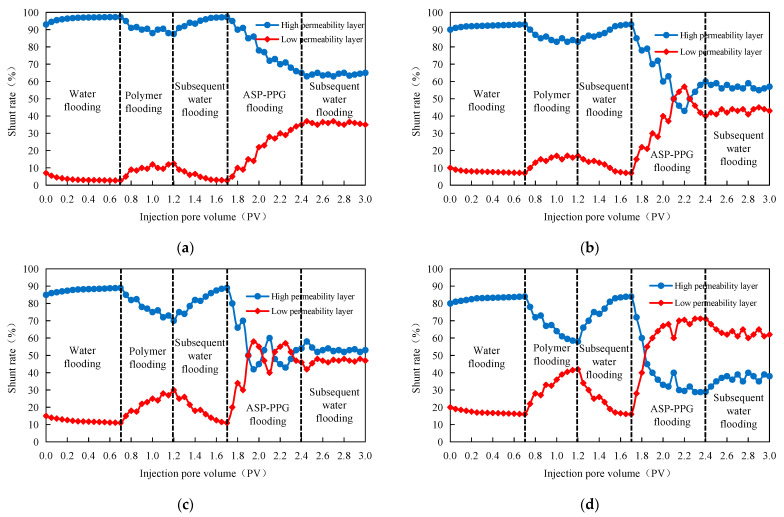
(**a**) The shunt rate curve of a two-pipe parallel experiment with a permeability level difference of 15. (**b**) The shunt rate curve of the two-pipe parallel experiment with a permeability level difference of 9. (**c**) The shunt rate curve of a two-pipe parallel experiment with a permeability level difference of 6. (**d**) The shunt rate curve of the two-pipe parallel experiment with a permeability level difference of 3.6.

**Figure 10 polymers-15-03523-f010:**
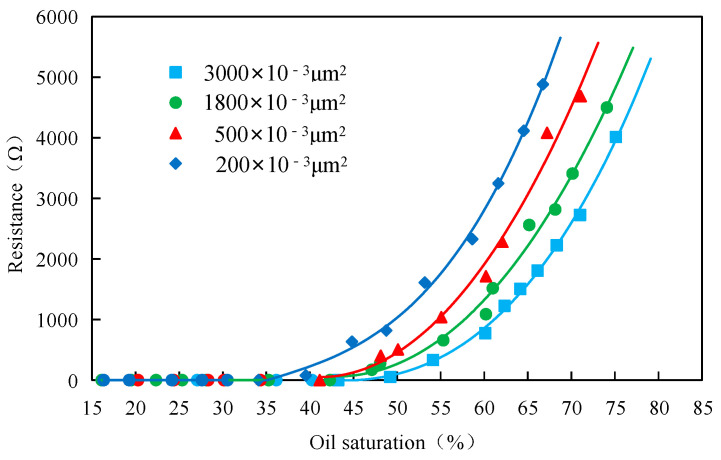
Resistance–oil saturation curve for different permeabilities.

**Figure 11 polymers-15-03523-f011:**
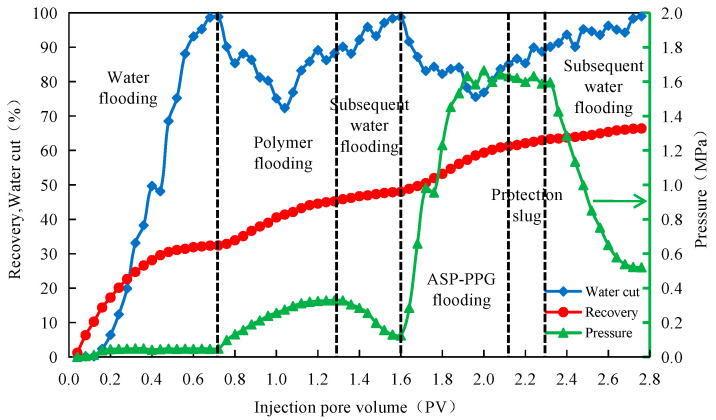
The curves of each parameter with the cumulative injected pore volume.

**Figure 12 polymers-15-03523-f012:**
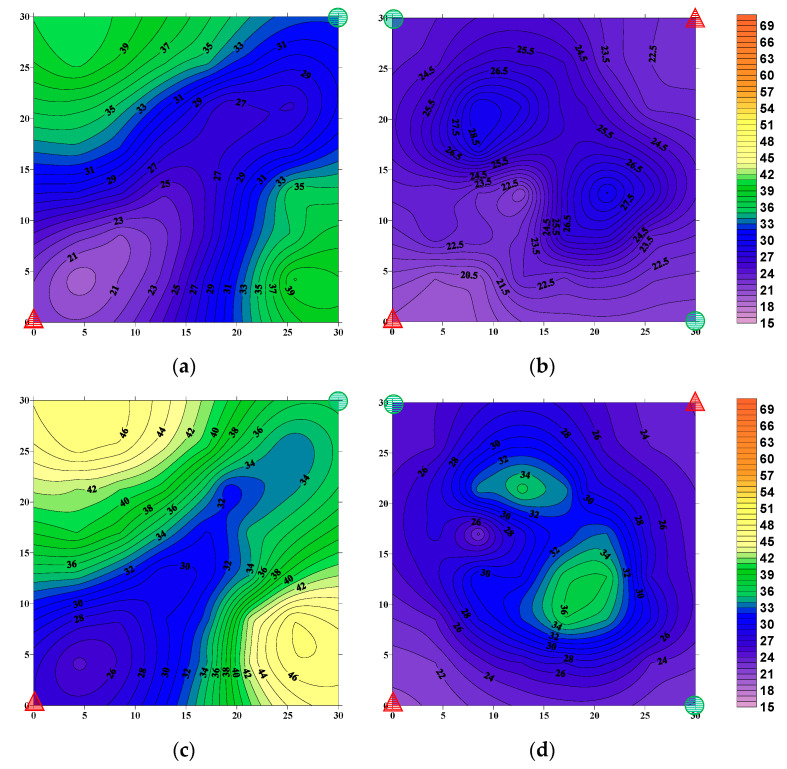
Changes in the oil saturation field in different permeability layers before and after the adaptive flooding, (**a**) before (**b**) after the adaptive system injection in the ultra-high permeability layer, (**c**) before the adaptive system injection in the high permeability layer, (**d**) after the adaptive system injection in the high permeability layer, (**e**) before the adaptive system injection, (**f**) after the adaptive system injection, (**g**) before the adaptive system injection in the low permeability layer, and (**h**) after the adaptive system injection in the low permeability layer. (The triangle is the injection well and the circle is the production well).

**Table 1 polymers-15-03523-t001:** Reagent and dosage used for the simulation formation water with a salinity of 6778 mg/L.

	NaHCO_3_	NaCl	KCl	MgSO_4_	Na_2_SO_4_	CaCl_2_
Concentration (g/L)	2.829	3.489	0.02	0.262	0.114	0.064

**Table 2 polymers-15-03523-t002:** Specific parameters of two groups of porous seepage models.

No.	Simulated Layer	Permeability (×10^−3^ μm^2^)	Dimensions of Core (mm)	Porosity (%)
1	High permeability layer	1800	1000 × 45 × 45	27.02
2	Low permeability layer	3000		28.89

**Table 3 polymers-15-03523-t003:** Specific parameters of four groups of two-pipe parallel models.

No.	Permeability (×10^−3^ μm^2^)	Dimensions of Core (mm)	Porosity (%)	Permeability Difference
1	200	300 × 45 × 22.5	22.68	15
	3000		28.72	
2	200	300 × 45 × 22.5	22.67	9
	1800		26.13	
3	500	300 × 45 × 22.5	23.23	6
	3000		28.71	
4	500	300 × 45 × 22.5	23.22	3.6
	1800		26.12	

**Table 4 polymers-15-03523-t004:** Specific parameters of the four-layer heterogeneous core well pattern model.

Layers	Permeability (×10^−3^ μm^2^)	Dimensions of Core (mm)	Porosity (%)	Mean Porosity (%)
Low permeability layer	200	300 × 300 × 15	22.58	25.17
Intermediate permeability layer	500		23.21	
High permeability layer	1800		26.11	
Ultra-hypertonic layer	3000		28.79	

**Table 5 polymers-15-03523-t005:** The pressure difference of each section of the long core, the resistance coefficient FR, and the residual resistance coefficient FRR of the system.

Permeability (×10^−3^ μm^2^)	Pressure Measuring Point	Water Flooding Stage (MPa)	ASP-PPG System (MPa)	Subsequent Water Flooding (MPa)	Drag Coefficient	Residual Drag Coefficient
1800	Pressure 0	0.006	0.85	0.31	141.7	51.7
	Pressure 1	0.006	0.61	0.24	101.7	40.0
	Pressure 2	0.006	0.41	0.19	68.3	31.7
	Pressure 3	0.006	0.29	0.16	48.3	26.7
	Pressure 4	0.006	0.18	0.12	30.0	20.0
3000	Pressure 0	0.004	0.52	0.16	130.0	40.0
	Pressure 1	0.004	0.34	0.12	85.0	30.0
	Pressure 2	0.004	0.26	0.09	65.0	22.5
	Pressure 3	0.004	0.18	0.07	45.0	17.5
	Pressure 4	0.004	0.11	0.05	27.5	12.5

**Table 6 polymers-15-03523-t006:** Results of the profile improvement rate.

No.	Permeability (×10^−3^ μm^2^)	Permeability Level Difference	Relative Suction Ratio (%)		Profile Improvement Rate (%)
			Before Treatment	After Treatment	
1	3000	15	97.3	65	94.85
	200		2.7	35	
2	1800	9	93	57	90.02
	200		7	43	
3	3000	6	89	53	86.06
	500		11	47	
4	1800	3.6	84	38	88.33
	500		16	62	

**Table 7 polymers-15-03523-t007:** Experimental results of adaptive flooding with the two-injection and two-extraction pattern after polymer flooding.

Recovery Factor in Water Flooding Stage (%)	Recovery Factor in Polymer Flooding Stage (%)	Recovery Factor in Adaptive Flooding Stage (%)	Total Recovery Rate (%)
32.4	15.6	18.4	66.4

## Data Availability

Data available on request due to restrictions, e.g., privacy or ethical. The data presented in this study are available on request from the corresponding author. The data are not publicly available due to company requirements.

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
