# Peer review of "Experimental Study on Enhanced Oil Recovery of Adaptive System after Polymer Flooding"

_polymers, 2023, doi:10.3390/polym15173523_

Round 1
Reviewer 1 Report
The manuscript entitled “Experimental study on enhanced oil recovery of adaptive system after polymer flooding” should be an interesting paper to the readers. This work showed the gelation and plugging application by injecting PPG with ASP system. In this manuscript, Pi et al. have formed adaptive oil displacement system by combining with ASP system. They evaluated the PPG performance in the heterogenous rock with their system by making three permeability layer system with good permeability contrast.
The results merit to publish in “gels” journal but I am not sure whether it is right to publish in Polymers. Here the main idea and work is on PPG gel not in polymer. This work can be done without ASP flood. Overall, the paper provides enough work for the paper. I would recommend its publication after the following minor questions are properly addressed in the revised manuscript.
1. Why the ASP has to be flooded? Could not you conduct SP system?
2. Information on experimental part such as how PPG were dispersed and injected? As they are big particles of micron size how they get injected in the core? What is the density of the particles? What is the condition of un-swelling and swelling? Need to explain in details
3. What is PPG? Is this preformed particle gel? Need to write the correct name.
4. Page 1, line 14, 15 needs to rewrite. What is deep core here?
5. Page 2, line 35, 36. Why the reservoir would be more heterogeneous after polymer flooding?
6. In page 1, line 44, page 2, lines 46, 48, 50 and in many lines and other pages, authors have written the reference number after author last name and before et al. which is not proper way of writing it. Need correction. The correct way is in page 2, line 53, reference number 11.
7. page 3, line 128, core flow experiment should be core flood experiment
8. PPG gel particles are of 0.15-0.3 mm size. They are synthesized as it is or crushed to the size as mentioned. How many times they swell up and under what condition? Need explanation
9. Why flow rate is very high of 0.5ml/min?
10. In Figure 6, IFT is shown in secondary Y-axis and viscosity in Primary Y axis. It is better to show with arrow to the curve towards right axis.
11. Figure 11 is also similar to figure 6. Only pressure is shown in secondary Y axis so please show with arrow to the pressure curve towards secondary Y axis.
The quality of English is acceptable but not great!
Reviewer 2 Report
The mauscript provides a concise overview of the study, mentioning the challenges of polymer flooding in Daqing Oilfield and the development of an adaptive oil displacement system (ASP-PPG) to address these challenges. The mauscript outlines the evaluation of the system's performance and the experimental results obtained. However, the mauscript could benefit from some improvements in terms of clarity and completeness.
Context and Objectives: The mauscript does not provide a clear statement of the research objectives. It should include a sentence explaining the specific goal of the study, such as "This study aims to evaluate the effectiveness of an adaptive oil displacement system (ASP-PPG) in improving the recovery efficiency of heterogeneous reservoirs after polymer flooding."
Methodology: The manuscript briefly mentions the evaluation of the system's performance and relevant experiments carried out. It would be helpful to provide a sentence listing the main experimental methods used, such as "The evaluation included static performance tests, flow experiments, microscopic experiments, profile control experiments, and four-layer heterogeneous physical model flooding experiments."
Specific Findings: While the manuscript presents some experimental results, it lacks specific data or percentage differences to highlight the significance of the findings. For example, it could include specific values for the improvement in recovery degree and the decrease in oil saturation in different layers.
Impact: The manuscript does not explicitly discuss the implications or potential impact of the findings. It would be beneficial to include a sentence or two discussing how the results contribute to addressing the challenges of polymer flooding and their potential significance for the oil and gas industry.
Conclusion: The manuscript could be concluded more effectively by summarizing the key findings and their implications. For example, "In conclusion, the ASP-PPG flooding system demonstrated enhanced stability, mobility, and profile improvement capability, resulting in an 18.4% improvement in recovery degree after polymer flooding. The adaptive system effectively utilized remaining oil in low and medium permeability layers, showing promise for improving recovery rates in challenging reservoir conditions."
The mauscript provides a concise overview of the study, mentioning the challenges of polymer flooding in Daqing Oilfield and the development of an adaptive oil displacement system (ASP-PPG) to address these challenges. The mauscript outlines the evaluation of the system's performance and the experimental results obtained. However, the mauscript could benefit from some improvements in terms of clarity and completeness.
Context and Objectives: The mauscript does not provide a clear statement of the research objectives. It should include a sentence explaining the specific goal of the study, such as "This study aims to evaluate the effectiveness of an adaptive oil displacement system (ASP-PPG) in improving the recovery efficiency of heterogeneous reservoirs after polymer flooding."
Methodology: The manuscript briefly mentions the evaluation of the system's performance and relevant experiments carried out. It would be helpful to provide a sentence listing the main experimental methods used, such as "The evaluation included static performance tests, flow experiments, microscopic experiments, profile control experiments, and four-layer heterogeneous physical model flooding experiments."
Specific Findings: While the manuscript presents some experimental results, it lacks specific data or percentage differences to highlight the significance of the findings. For example, it could include specific values for the improvement in recovery degree and the decrease in oil saturation in different layers.
Impact: The manuscript does not explicitly discuss the implications or potential impact of the findings. It would be beneficial to include a sentence or two discussing how the results contribute to addressing the challenges of polymer flooding and their potential significance for the oil and gas industry.
Conclusion: The manuscript could be concluded more effectively by summarizing the key findings and their implications. For example, "In conclusion, the ASP-PPG flooding system demonstrated enhanced stability, mobility, and profile improvement capability, resulting in an 18.4% improvement in recovery degree after polymer flooding. The adaptive system effectively utilized remaining oil in low and medium permeability layers, showing promise for improving recovery rates in challenging reservoir conditions."
Round 2
Reviewer 2 Report
Accept